# Synthesis and Biological Evaluation of Lipophilic Nucleoside Analogues as Inhibitors of Aminoacyl-tRNA Synthetases

**DOI:** 10.3390/antibiotics8040180

**Published:** 2019-10-09

**Authors:** Manesh Nautiyal, Bharat Gadakh, Steff De Graef, Luping Pang, Masroor Khan, Yi Xun, Jef Rozenski, Arthur Van Aerschot

**Affiliations:** 1KU Leuven, Rega Institute for Medical Research, Medicinal Chemistry, Herestraat 49 BOX 1030, 3000 Leuven, Belgium; manesh2006@gmail.com (M.N.); bkgadakh@gmail.com (B.G.); luping.pang@kuleuven.be (L.P.); mkhan@ibch.poznan.pl (M.K.); yixun13@gmail.com (Y.X.); jef.rozenski@kuleuven.be (J.R.); 2KU Leuven, Department of Pharmaceutical and Pharmacological Sciences, Biocrystallography, Herestraat 49 BOX 822, 3000 Leuven, Belgium; steff.degraef@kuleuven.be

**Keywords:** aminoacyl-tRNA synthetase, aminoacylated sulfamoyladenosines, bisubstrate competitive inhibitor, prodrugs, lipophilic adenosines, enzyme inhibition, antibacterial activity

## Abstract

Emerging antibiotic resistance in pathogenic bacteria and reduction of compounds in the existing antibiotics discovery pipeline is the most critical concern for healthcare professionals. A potential solution aims to explore new or existing targets/compounds. Inhibition of bacterial aminoacyl-tRNA synthetase (aaRSs) could be one such target for the development of antibiotics. The aaRSs are a group of enzymes that catalyze the transfer of an amino acid to their cognate tRNA and therefore play a pivotal role in translation. Thus, selective inhibition of these enzymes could be detrimental to microbes. The 5′-*O*-(*N*-(L-aminoacyl)) sulfamoyladenosines (aaSAs) are potent inhibitors of the respective aaRSs, however due to their polarity and charged nature they cannot cross the bacterial membranes. In this work, we increased the lipophilicity of these existing aaSAs in an effort to promote their penetration through the bacterial membrane. Two strategies were followed, either attaching a (permanent) alkyl moiety at the adenine ring via alkylation of the *N*^6^-position or introducing a lipophilic biodegradable prodrug moiety at the alpha-terminal amine, totaling eight new aaSA analogues. All synthesized compounds were evaluated in vitro using either a purified *Escherichia*
*coli* aaRS enzyme or in presence of total cellular extract obtained from *E. coli.* The prodrugs showed comparable inhibitory activity to the parent aaSA analogues, indicating metabolic activation in cellular extracts, but had little effect on bacteria. During evaluation of the *N*^6^-alkylated compounds against different microbes, the *N*^6^-octyl containing congener **6b** showed minimum inhibitory concentration (MIC) of 12.5 µM against *Sarcina lutea* while the dodecyl analogue **6c** displayed MIC of 6.25 µM against *Candida*
*albicans*.

## 1. Introduction

The surge of antimicrobial-resistant strains of pathogenic bacteria is a significant concern to human health worldwide and has a profound impact on hospital born infections, chemotherapy, tuberculosis, and surgical procedures. This resistance crisis is estimated to cause 300 million cumulative premature deaths by 2050, with a loss of up to $100 trillion to the global economy [1]. A recent report from Public Health England stated that antibiotic resistance could make three million surgical procedures deadly in England alone [2]. The situation becomes more critical as there are very few antibiotics in the drug discovery pipeline [3]. To overcome antimicrobial resistance, various government organizations, academic institutions, and industry specialists have made different suggestions ranging from prevention of infection [4] to rational use of antibiotics, over the use of alternatives (e.g., vaccines, bacteriophages, etc.) [5], for the development of faster diagnostic methods or new techniques of diagnosis. In addition, the combination of antibiotics or the modification of existing antibiotics, and finally, finding completely new antibiotics directed towards novel targets for which development of resistance will be less straightforward [6,7] are alternative strategies in our continuous war against microbes. As medicinal chemists, our focus is on the development of new classes of antibiotics against new and existing targets, which must be essential for bacterial cell survival.

The major problem with the discovery of target-based antibiotics is the failure of compounds in an early phase of development due to their inability to permeate through the bacterial membrane. The bacterial membranes are complex and vary in different microbes. Based on the differences in the composition of cell membrane bacteria are divided into two classes, Gram-positive and Gram-negative. The former has a thick cell wall, composed of multiple layers of peptidoglycan and anionic glycopolymers called teichoic acids, which allow passage of nutrients and small molecules. However, Gram-negative bacteria have comparatively thin cell walls, composed of only a few layers of peptidoglycans surrounded by an outer membrane containing lipopolysaccharides. The cell-envelope permeability barrier is particularly strong in pathogens like *Pseudomonas aeruginosa*, *Burkholderia cepacia*, and some other Gram-negative bacteria [8]. These species are mainly associated with pneumonia, bacteremia, and lung infection in cystic fibrosis. Many antimicrobials proved ineffective against Gram-negative bacteria, presumably because they could not cross the cell membrane and did not reach the site of action. Due to this varying difference in bacterial membranes compared to eukaryotic cells, it can be inferred that structural and physicochemical properties that govern compound permeability (i.e., the Lipinski rule of five [9]) in eukaryotic cells will not be applicable in antibiotics. The high molecular weight and increased polarity in antibiotics are the most notable physicochemical property differences in comparison to other drug classes [10]. However, due to significant differences in the bacterial membrane between the different microbial groups, it is difficult to envision the set of rules governing the passage of compounds through a bacterial membrane.

In the current work, we aim for the development of in vivo active 5′-*O*-(*N*-(L-aminoacyl)) sulfamoyladenosines (aaSAs) based antibiotics. aaSAs are the most potent inhibitors of the corresponding aminoacyl-tRNA synthetases (aaRSs). aaRSs catalyze ligation of the correct amino acid to their cognate tRNA and therefore play a vital role in determining fidelity and accuracy of protein synthesis [11]. aaSAs are mimics of the reaction intermediate aminoacyl-adenylate (aa-AMP) (Figure 1, Structure a) and act by competitively inhibiting aaRSs, thus preventing the translation, and ultimately causing cell death. Although aaSAs are nanomolar bisubstrate inhibitors of aaRSs, they are inactive against bacteria due to their poor permeability. Recently Davis et al. analyzed the various physicochemical properties of aaSA (targeting adenylation enzymes), which are responsible for the permeation through bacterial membranes [12]. From this study, it was observed that aaSA analogues having high logP values (high lipophilicity) accumulated more in bacteria as compared to analogues having low logP values. Therefore, while the authors proved many more physicochemical parameters are influencing uptake at varying extent across different bacteria, the inactivity of aaSAs (targeting aaRSs) can be correlated with their high hydrophilicity [12].

There are several strategies to increase the antimicrobial activity of compounds suffering from limited permeability through the bacterial membrane. One of them is the combination of the antibiotic (inhibitor) with a compound acting on the outer membrane of the bacterial cell wall altering its permeability, and leading to increased concentration of the antibiotic (inhibitor) in the bacterial cells via increased diffusion. 

Examples of compounds acting on the outer membrane include pore-forming antimicrobial lipopeptides like daptomycin [13]. An alternative approach could be the conjugation of the inhibitor with a transport vector like siderophores or peptide uptake signals and make use of the associated transport system, i.e., the “Trojan-horse” strategy, as seen for instance with microcin C [14]. and albomycin [15,16]. In the past, our group has tried such a Trojan-horse approach to promote the uptake of aaSA analogues and provide antibacterial activity without success [17,18,19,20,21]. An alternative strategy to increase the permeability of compounds is based on increasing the lipophilicity, especially for polar compounds. This could be accomplished by the addition of a lipophilic moiety to the lead compound at a non-interacting or least interacting site, or by introducing a lipophilic cleavable moiety, to release the active warhead following uptake of the “disguised” compound.

## 2. Design Rationale

In this work, we tried both the above-mentioned approaches to increase the lipophilicity of aaSAs, which should lead to an increased influx of compounds and antibacterial activity. The first approach involved the coupling of different lipophilic groups to the *N^6^*-amino group of aaSA (Figure 1). As observed from the crystal structures of *Thermus thermophilus* (Tt) isoleucine tRNA synthetase (IleRS) in complex with isoleucylsulfamoyl adenosine (ISA) or with mupirocin (ileRS inhibitor; unpublished work), there is sufficient space available around the adenine base to accommodate some chemical modifications [22,23]. In addition, the N^6^-amine is only making one hydrogen bond interaction with the backbone of the protein chain. Likewise, the aliphatic chain of mupirocin is interacting with the adenine binding domain [24]. strengthening our choice for longer alkyl moieties. Therefore, we alkylated ISA at the *N*^6^-position with alkyl moieties of various chain length to analyze for the best fit within the cavity and to achieve the maximum inhibitory effect. Thus, various chain lengths from octyl, dodecyl to octadecyl, and a phenyl group were introduced at the N^6^-amino moiety of the adenosine part (**6b–e**). Additionally, the 6-NH-CH_3_ (**6a**) and 6-*O*-methyl moieties (**6f**) were introduced to evaluate the effect of oxygen vs nitrogen regarding enzymatic affinity, as well as the orientation of the methyl substituent in the active site of ileRS.

The second approach deals with the attachment of a promoiety at the α-amine of aaSA (Figure 1, structure b). Two carbamate promoieties which have been shown to be cleaved intracellularly following uptake, are the 4-nitrobenzyloxycarbonyl [25,26] and 4-acetoxybenzyloxycarbonyl carbamate [27,28]. The functional group at the para-position (either nitro or acetoxy here) on activation by nitroreductase or esterase respectively, should lead to cleavage of the prodrug and release of the active aaSA. The mentioned functionalities therefore are attached at the α-amine of leucylsulfamoyl adenosine (LSA) providing compounds **14** and **18** as the desired prodrugs. Following synthesis, their biological activity will be measured against different microorganisms while their intrinsic activity will be tested in a cellular extract using the aminoacylation assay. For this prodrug synthesis, we opted for leucylsulfamoyl adenosine, which already showed to be strongly active in vitro against leucine tRNA synthetase (LeuRS).

The first mentioned activation calls for the nitroreductase (NTR) [20], which are not found in human for activation of **14**, and hence could provide a selectivity handle. In fact, the nitrobenzyl moiety has been used, as well as a selectivity handle, in cancer treatments in generating directed prodrugs with a metabolic trigger, but then requires the NTR to be introduced as well by transfection methodologies or other means [29]. The second principle makes use of various esterases to release LSA from **18**. Both enzyme types are present in several bacteria such as *E. coli*. Following exposure to the enzyme, the inactive prodrug should be metabolized to produce the active compound. In a typical prodrug strategy, the 4-nitrobenzyl group is attached to a leaving moiety such as a phosphoramide or a carboxylate. We here opted for the 4-nitrobenzyloxycarbonyl which on reduction by NTR produces the *p*-hydroxylamino-benzyl carbamate. The latter is prone to hydrolytic cleavage (Scheme 1) as the transformation converts an electron-withdrawing nitro group into an electron donating group [25,26]. Likewise, esterases upon hydrolysis of **18** will provide the hydrolytically cleavable *p*-hydroxybenzylcarbamate leading to the active compound [27] (Scheme 1).

## 3. Results

### 3.1. Chemistry

For the synthesis of *N*^6^-alkylated compounds, initially, we attempted to introduce various *N*^6^-alkyl groups at the pre-final protected stage by nucleophilic aromatic substitution on the C^6^-position. However, we observed no reaction or degradation of the protected 6-chloropurine ISA analogue. Later we attempted to modify the 6-chloropurine analogue of sulfamoyl adenosine; however, when trying to substitute 6-Cl with different alkylamines on **3f**, cleavage of the sulfamoyl group was observed, as of the presence of the base *N,N*-diisopropylethylamine (DIPEA) and quite harsh microwave conditions (Appendix A). We finally opted for introduction of the various alkylamines on the protected 6-chloropurine riboside (**2**), followed by sulfamoylation and amino acid coupling to obtain the respective coupled products **5a**–**f** which on deprotection gave compounds **6a**–**f**.

#### 3.1.1. Synthesis of N^6^–alkylated Analogues of 5′-*O*-(*N*-(L-isoleucyl)) Sulfamoyl Adenosine

As shown in Scheme 2**,** previously reported strategies were followed to obtain the desired *N^6^*-alkylated derivatives. Their synthesis started via acetonide protection of commercially available 6-chloropurine riboside (**1**) utilizing dimethoxypropane using para-toluene sulfonic acid as the catalyst and acetone as solvent. The acetonide-protected **2** on microwave-assisted nucleophilic aromatic substitution with a series of alkylamines and aniline afforded compounds **3a**–**e** [30].

In the next reaction, these compounds **3(a–e)** were sulfamoylated at the 5′-*O*-position using in situ prepared sulfamoyl chloride, which is synthesized by reacting formic acid with chlorosulfonylisocyanate [20], to obtain **4(a–e)**. These were then coupled with the *N*-hydroxysuccinimide active ester of Boc-Ile (Boc-Ile-OSu) in the presence of DBU, leading to the formation of the coupled products **5(a–e)**. Further deprotection of boc and acetonide functionality by the action of 60% TFA:water mixture lead to the formation of the desired compounds **6(a–e)**. For the synthesis of the *O*^6^-methylated analogue, 5′-*O* sulfamoylation of acetonide-protected 6-chloropurine riboside **2,** was performed generating compound **3f**. Next, this compound on nucleophilic aromatic substitution by sodium methoxide in methanol [31]. yielded compound **4f,** which on further coupling with Boc-Ile-OSu and deprotection using 60% TFA: H_2_O mixture led to the desired **6f**.

#### 3.1.2. Synthesis of α-amine Promoiety Analogues

The synthesis of the intended prodrugs was carried out based on reported methodologies [30] and are described in Scheme 3 and Scheme 4. Synthesis started from the commercially available adenosine **7**, which was first persilylated to generate compound **8** after which the 5′-position was liberated using a mixture of TFA:THF:H_2_O. The obtained **9** on reaction with the in situ generated sulfamoyl chloride led to the formation of **10**. The sulfamoylated adenosine **10** was coupled with Boc-Leu-OSu in the presence of DBU to yield **11**. The coupled product **11** was then treated with a 50:50 mixture of TFA and DCM to yield **12**. The 4-nitrobenzyl chloroformate was reacted by using DIPEA as base and DMF as the solvent. The promoiety coupled compound **13** is finally treated with HF in TEA to cleave the silyl protection which lead to the desired prodrug **14**.

For the synthesis of 4-acetoxybenzyloxycarbonyl protected α-amine, the 4-acetoxybenzyl chloroformate **16** was generated in situ by reacting triphosgene with 4-acetoxybenzyl alcohol **15** in the presence of DIPEA [32]. The 4-acetoxybenzyl chloroformate is used as such for further coupling with compound **12** which led to the silyl protected intermediate **17** that on further treatment with HF in TEA yielded the desired compound **18.**

### 3.2. Biology

All synthesized compounds were evaluated in vitro against either purified enzyme (IleRS) in a buffer or in a cellular extract (S30), followed by determination of MIC values against different microbes. 

#### 3.2.1. Measurement of in vitro Inhibitory Activity with Purified *E. coli* IleRS

Following successful synthesis of compounds (**6a–f**), inhibitory activity was determined using radiolabel transfer assay. IleRS and total tRNA isolated from *E. coli* were used. In this assay the quantity of C^14^ labelled isoleucine transferred to tRNA^Ile^ was determined by precipitating the [C^14^] Ile-tRNA^Ile^ complex using a 10% TCA solution. Out of six tested derivatives, four showed IC_50_ in the nanomolar range in the radiolabel transfer assay (**6a,b and 6e,f**; Figure 2A and Table 1**)**. However, for all four a 2- or 3-fold decrease in inhibitory activity was observed versus ISA in analogy with which was reported in the past [20,33]. A general trend of decrease in inhibitory activity was observed with increasing alkyl chain length. 

Indeed, the octadecyl compound **6d** was found to be active only in the micromolar range, while the dodecyl derivative **6c** was active in submicromolar range (Figure 2B); in view of their lower inhibitory activity, their dose-response curves were not determined. Remarkably, the *O*^6^-methyl derivative **6f** showed about 3-fold better inhibitory activity compared to the *N*^6^-methylated ISA **6a** and matched the activity of ISA (Table 1), while the phenyl substituted ISA (**6e**) showed activity similar to methylamine substituted ISA (**6a**). 

#### 3.2.2. Time-dependent in vitro Inhibitory Activity with *E. coli* Cellular Extract

The compounds with either a 4-nitrobenzyloxycarbonyl (**14**) or 4-acetoxybenzyloxycarbonyl moiety (**18**) attached to the alpha-amino group of LSA were tested in *E. coli* S30 cellular extract to determine the time required for metabolic activation of the mentioned promoieties. Therefore, the compounds were incubated with the cellular extract at 37 °C for different time periods and inhibitory activities were measured in comparison with LSA. The cellular lysate of *E. coli* appeared to be enriched with the nitroreductases and esterases responsible for activation to the parent compound. Both prodrugs showed equal inhibitory effect on LeuRS compared to LSA, irrespective of the time of incubation. Hence, in view of using only 250 nM of prodrug equivalent to the concentration of the parent inhibitor LSA, the (partial) early release of the LSA warhead must be concluded (Figure 3).

#### 3.2.3. Antimicrobial Assay

All the synthesized analogues were tested against six different microbes to cover the spectrum of activity against gram-positive—*Staphylococcus aureus* ATCC 6538P, *Staphylococcus epidermidis* RP62A, *Sarcina lutea* ATCC9341; gram-negative—*Escherichia*
*coli* NCIB 8743, *P. aeruginosa* PAO1; and fungi—*Candida albicans* CO11. The results are described in Table 2. The antimicrobial activities were obtained by measuring the optical density at 600 nM of the individual wells of a microtiter plate reached by the microbial culture in the presence of different concentrations of respective inhibitors.

The aaSA derivatives having a dodecyl or octyl substitution at the *N^6^*-position proved to be active against some microbes. The octyl derivative (**6b**) showed the best MIC against *S. lutea* of 12.5 µM and similarly, dodecyl derivative (**6c**) showed the best MIC of 6.25 µM against *C. albicans.* This proves an amelioration of the uptake properties via increase of hydrophobic-hydrophilic balance, but less than was hoped for. Potentially, the distance between hydrophobic and hydrophilic portions of the molecule or so called “amphiphilic moment”, likewise contributes to improved uptake properties [35]. Unexpectedly, both compounds with a prodrug moiety were found to be inactive under the applied assay conditions. Possibly the promoieties are released too rapidly to secure sufficient uptake of these prodrugs. The compound with octadecyl (**6d**) substitution on the other hand, might be too lipophilic to dissolve thoroughly in the culture media (providing aggregates) or proved unable to cross the bacterial membrane to show its activity.

#### 3.2.4. Computational Analysis of Molecular Properties

For understanding the change in physicochemical properties of the synthesized compounds with respect to the parent analogues, computational prediction of molecular properties was attempted. The different parameters were determined using the online physicochemical predictor toolkit (www.molinspiration.com/cgi-bin/properties). Both modification strategies as expected appeared to be increasing the compounds’ partition coefficient (logP). The analogues carrying either a phenyl, octyl, or dodecyl substitution at the *N^6^*–position and both synthesized prodrugs more or less satisfied the Lipinski rules (Table 3). Obviously, there is no change in the number of hydrogen bond donors or acceptors and in total polar surface area for C6-modified compounds. However, the number of hydrogen bond acceptors and polar surface area slightly increases for the synthesized prodrugs in comparison to LSA (not included in Table 3).

## 4. Discussion

The primary reason for the failure of antibacterial lead molecules discovered after a rational design approach by SAR, is their limited permeability through the bacterial membrane. In the past, trying to overcome the permeability issue of aaSA and their derivatives, our group already synthesized several conjugates comprising of aaSA analogues coupled with various transporter peptide [17,18,19,20,21] or siderophores [21]. However, we achieved limited success due to tedious synthesis, purification, and stability of peptide and siderophore coupled compounds.

There is widespread belief that the Lipinski “rule-of-five” guidelines for oral uptake of classical drugs should not be extrapolated to antibiotics and indeed many (natural) antibiotics do not comply with this guideline. A more recent report however, made a clear distinction between compounds targeting riboproteins or those targeting regular bacterial protein targets, where the latter upon analysis mostly comply with the Lipinski rules [36]. In addition, it has been observed that increase in lipophilicity of sulfamoyladenosine derivatives leads to increased permeability of the latter through the bacterial membrane [12]. Following our abovementioned approach, we were able to synthesize six compounds having lopP values from one to five obeying the Lipinski rule (Table 3**)**, except for the compounds having either a methoxy or octadecylamine substitution at the C6-position resulting in a too polar or too lipophilic compound, respectively.

Therefore, in aim to improve the in vivo efficacy of aaSAs, we used 2 different strategies utilizing either one of two amino functionalities present in aaSAs. As one can observe from crystal structures that there is considerable space around the adenine binding region [24], we decided to attach lipophilic moieties to the adenine heterocycle. Thus, we synthesized six lipophilic *N^6^*-modified analogues of ISA with the intention to find a compound which can cross the bacterial membrane while retaining the inhibitory activity. As expected, four compounds (**6a**, **6b**, **6e**, and **6f**) out of six were found to have quite similar inhibitory activity against IleRS as compared to the parent analogue ISA. The gradually decreasing inhibitory activity with increasing chain length (**6a–d**) could be due to increasing entropic losses of the longer alkyl moieties, insufficiently compensated by increased hydrophobic interaction. Compounds **6b** and **6c** showed some improved antimicrobial activity against different microbes, but less than was hoped for. The longer octadecyl chain presumably leads to aggregates, leading again to reduced uptake. These results also show that our approach of substituting the *N*^6^-position was the correct one, as only a limited reduction of the enzyme inhibitory activity was observed. This is in stark contrast to our previous efforts of methylating the alpha-amine, which was accompanied with a dramatic loss in inhibitory activity [37]. The adenine moiety seems not to be essential to generate high affinity molecules against class I aaRS, as it can be readily substituted with other nucleobases or even tetrazole moieties while still showing good inhibitory activity [11,38].

In the second approach, we used prodrug moieties to enhance the permeability and also increase the bacterial selectivity towards these compounds. Therefore, we opted for 4-acetoxybenzyloxycarbonyl and 4-nitrobenzyloxycarbonyl moieties, which can be cleaved by esterases or nitroreductases, respectively. Following coupling of the promoieties to the α-amine of the LSA, the compounds were evaluated in a cellular lysate. We hypothesized that compounds would take some time for activation and therefore planned a time-point study from 2 to 120 min. However, the compounds appeared to be sufficiently metabolized already within 2 min to retain the full activity of the parent molecules. On further testing of these prodrugs against different microbes, no significant inhibitory activity could be observed, which might hint to preliminary degradation in the LB media, or alternatively, still no uptake in bacterial cells is accomplished. Nitrobenzyl carbamates (NBC) of a variety of cytotoxic amines are metabolized efficiently by nitroreductases to the hydroxylamines, which fragment to release the amines [29,39,40]. Preliminary chemical hydrolysis of our prodrugs is unlikely at the physiological pH buffer conditions used (see experimental). This type of prodrug has been used before at many occasions, as a means of directed prodrugs for cancer treatment. NBC derivatives of doxorubicine with the carbamate attached to the aminated sugar daunosamine [29], resemble the closest our alpha-amine carbamate as in **14**, and still show 36% remaining carbamate following 24 h incubation in minimum essential medium (Eagle) supplemented with 5% fetal calf serum. Obviously however, selectivity in inhibiting the bacterial aaRS is another issue in development of the aaSA compounds, only dealt with in this work in using the nitrobenzylated prodrug. But the feasibility of selective targeting has been shown previously by the availability of two marketed drugs inhibiting an aaRS, with mupirocin (likewise equipped with a long aliphatic tail) and tavaborole, and in using heterocyclic sulfonamide scaffolds [41,42]. Our extensive 3D structural work [33] on various aaRS in complex with inhibitory ligands (including unpublished work) will further pave the way for improved selectivity.

## 5. Conclusions and Future Perspective

A challenging and lengthy synthesis of C^6^-purine substituted analogues of ISA and of two LSA based prodrugs was performed to increase the lipophilicity of their parent compounds. The C^6^-purine substituted compounds showed potent inhibitory activity versus purified IleRS in a radiolabel transfer assay, albeit at a slightly higher concentration than the parent compound. The synthesized prodrugs appeared very effective against LeuRS in an *E. coli* cellular extract, showing rapid conversion to the parent compound without apparent loss in efficacy. The compounds **6b** and **6c** proved most effective displaying MIC in the micromolar range against few microbes indicating a positive effect of octyl and dodecyl substitution on permeation through the bacterial membrane and subsequent IleRS inhibitory activity. In general, although we met some difficulties, we still believe these novel approaches for altering the physicochemical properties of potent but too polar lead molecules could be utilized for their further development towards a novel antibiotic targeting an aminoacyl-tRNA synthetase. Detailed crystallographic studies on the interaction of the *N^6^*-modified compound with IleRS will further guide the rational design of future compounds against aaRSs.

## 6. Experimental Section

### 6.1. Materials and Methods

Reagents and solvents were purchased from commercial suppliers and used as provided unless indicated otherwise. DMF and THF were of analytical grade and were stored over 4 Å molecular sieves. All other solvents used for reactions were analytical grade and used as provided. Reactions were carried out in oven-dried glassware under a nitrogen atmosphere with stirring at room temperature unless indicated otherwise. All microwave irradiation experiments were carried out in a dedicated CEM-Discover mono-mode microwave apparatus. C^14^-radiolabeled amino acids and scintillation liquid were purchased from Perkin Elmer. 

^1^H and ^13^C NMR spectra of the compounds dissolved in CDCl_3_, CD_3_OD, DMSO-*d_6_* or D_2_O have recorded on a Bruker Ultra Shield Avance 300 MHz, 500 MHz or when needed at 600 MHz spectrometers. The chemical shifts are expressed as δ values in parts per million (ppm), using the residual solvent peaks (CDCl_3_: ^1^H 7.26 ppm; ^13^C, 77.16 ppm; DMSO: ^1^H, 2.50 ppm; ^13^C, 39.52 ppm; HOD: ^1^H, 4.79 ppm; CD_3_OD: ^1^H, 3.31 ppm; ^13^C, 49.00 ppm) as a reference. Coupling constants are given in Hertz (Hz). The peak patterns are indicated by the following abbreviations: bs = broad singlet, d = doublet, m = multiplet, q = quadruplet, s = singlet and t = triplet. High-resolution mass spectra were recorded on a quadrupole time-of-flight mass spectrometer (Q-Tof-2, Micromass, Manchester, UK) equipped with a standard ESI interface; samples were infused in 2-propanol/H_2_O (1:1) at 3 µL.min^−1^. 

For TLC, precoated aluminium sheets were used (Merck, Silica gel 60 F_254_). The spots were visualized by UV light at 254 nm. Column chromatography was performed on ICN silica gel 60A 60–200 µm. Final products were purified using a C-18 110 Å column connected to a Shimadzu SPD-20A HPLC and Shimadzu SPD-20A detector. Eluent compositions are expressed as *v/v*. Recordings were performed at 254 nm and 214 nm. Analytical data are only provided for all new compounds.

### 6.2. Chemical synthesis of the Intermediates and Final Compounds

#### 6.2.1. 2,3′-isopropylidene 6-chloropurine Riboside (2)

As reported in literature [43], compound **1** (6-chloropurine riboside, 8 g, 0.029 mol) was stirred with a mixture of dimethoxypropane (DMP) (34.31 mL, 0.29 mol) and paratoluenesulfonic acid (PTSA) (2.66 g, 0.014 mol) in dry acetone (80 mL) at room temperature for overnight. Thin layer chromatography TLC (developed at 10% methanol in dichloromethane (DCM)) was used to monitor the reaction. Saturated sodium bicarbonate was added to quench the reaction. Afterwards, the solvent was evaporated under reduced pressure. The crude product was dissolved in DCM, and the organic layer was washed two times with saturated sodium bicarbonate and one time with brine. Column chromatography was performed with a gradient of 5% methanol in DCM to obtain the desired compound **2** at 86% yield. ^1^H NMR (300 MHz, CDCl_3_) δ 1.37 (s, 3H, C–CH_3_), 1.64 (s, 3H, C–CH_3_), 3.81 (m, 1H, H-5′a), 3.97 (m, 1H, H-5′b), 4.54 (m, 1H, H-4′), 5.00 (dd, *J* = 10.0, 2.6 Hz, 1H, H-3′), 5.10 (dd, *J* = 6.0, 1.6 Hz, 1H, H-2′), 5.19 (dd, *J* = 6.0, 4.5 Hz, 1H, H-1′), 6.00 (d, *J* = 4.5 Hz, 1H, 5′OH ), 8.28 (s, 1H, H-2), 8.75 (s, 1H, H-8). ^13^C NMR (75 MHz, CDCl_3_) δ 25.5 (C–CH_3_), 27.8 (C–CH_3_), 63.4 (C-5′), 81.8 (C-4′), 83.7 (C-3′), 86.8 (C-2′), 94.2 (C-1′), 114.8 (C–(CH3)_2_), 145.0 (C-5), 152.0 (C-4), 152.5 (C-2). HRMS [ESI] *m/z*: calcd. for C_13_H_16_ClN_4_O_4_ ([M+H]^+^) 327.0854, found: 327.0835.

#### 6.2.2. 2′,3′-isopropylidene-N^6^-(methyl)-adenosine (3a)

Compound **2** (200 mg, 0.613 mmol) was dissolved in 6 mL of DMSO in a microwave vial. Methylamine (40% solution in water) (1.5 equiv, 0.102 mL) and DIPEA (3 equiv, 0.32 mL, 1.84 mmol) were added, and the mixture was put in a microwave for 45 min, at 110 °C utilizing 150W power. The reaction was monitored by doing TLC in an acetone/hexane gradient (50:50 *v/v*). To perform the TLC, take 100 microliters of the crude reaction mixture and add 500 microliters of ethyl acetate and 500 microliters of water to an Eppendorf tube. Extract the organic layer and use this for TLC. After complete conversion, 350 mL of DCM was added, and the organic layer was washed with 100 mL potassium hydrogen sulfate (10% m/v) and 100 mL of brine. The organic layer was evaporated under vacuum, and column chromatography was performed with a gradient of 40% acetone in hexane to obtain **3a**. Yield: 97% (0.19 g). ^1^H NMR (300 MHz, CDCl_3_) δ 1.31 (s, 3H, C–CH_3_), 1.58 (s, 3H, C–CH_3_), 3.10 (s, 3H, N^6^–CH_3_), 3.74 (d, *J* = 12.8 Hz, 1H, H-5′a), 3.91 (dd, *J* = 12.9, 1.6 Hz, 1H, H-5′b), 4.48 (s, 1H, H-4′), 5.06 (dd, *J* = 5.7, 1.3 Hz, 1H, H-3′), 5.17 (t, 1H, H-2′), 5.81 (d, *J* = 4.8 Hz, 1H, H-1′), 6.53 (s, 1H, N^6^-H), 6.75 (bs, 1H, 5′OH), 7.74 (s, 1H, H-2), 8.28 (s, 1H, H-8). ^13^C NMR (75 MHz, CDCl_3_) δ 25.5 (C-CH_3_), 27.9 (C–CH_3_), 29.6 (N^6^-CH_3_), 63.6 (C-5′), 82.0 (C-4′), 83.4 (C-3′), 86.5 (C-2′), 94.4 (C-1′), 114.2 (C-5), 121.6 (C–(CH_3_)_2_), 139.6 (C-8), 153.1 (C-2), 156.1 (C-6). HRMS [ESI] m/z: calcd. for C_14_H_20_N_5_O_4_ ([M+H]^+^) 322.1510, found: 322.1512.

#### 6.2.3. 2′,3′-isopropylidene-N^6^-(octyl)-adenosine (3b)

Compound **2** (200 mg, 0.613 mmol) was added to a microwave vial containing octylamine (0.152 mL, 0.92 mmol) and DIPEA (0.32 mL, 1.84 mmol) in DMSO (6 mL). The reaction condition, TLC and purification were similar as **3a**. Yield: 90% (0.23 g). ^1^H NMR (300 MHz, CDCl_3_) δ 1.21–1.64 (m, 23H, C–(CH_3_)_2_, H_3_C–(CH_2_)_7_), 3.56 (bs, 2H, N^6^–CH_2_), 3.74 (dd, *J* = 12.7, 1.6 Hz, 1H, H-5′a), 4.90–3.95 (m, 1H, H-5′b), 5.06 (dd, *J* = 5.9, 1.2 Hz, 1H, H-4′), 5.17 (t, 1H, H-3′), 5.81 (d, *J* = 4.8 Hz, 1H, H-2′), 6.11 (t, 1H, H-1′), 6.78 (d, *J* = 11.0 Hz, 1H, 5′OH), 7.73 (s, 1H, H-2), 8.28 (s, 1H, H-8). ^13^C NMR (75 MHz, CDCl_3_) δ 14.1- 40.9 (C–(CH_3_)_2_, H_3_C–(CH_2_)_7_), 63.6 (C-5′), 82.0 (C-4′), 83.4 (C-3′), 86.5 (C-2′), 94.5 (C-1′), 114.2 (C-5), 121.4 (C–(CH_3_)_2_), 139.6 (C-8), 153.1 (C-2), 155.6 (C-6). HRMS [ESI] m/z: calcd. for C_21_H_34_N_5_O_4_ ([M+H]^+^) 420.2605, found: 420.2597.

#### 6.2.4. 2′,3′-isopropylidene-N^6^-(dodeceyl)-adenosine (3c)

Compound **2** (200 mg, 0.613 mmol) was added to a microwave vial containing dodecylamine (170.43 mg, 0.92 mmol) and DIPEA (0.32 mL, 1.84 mmol) in DMSO (6 mL). The reaction condition, TLC and purification were similar as **3a**. Yield: 71% (0.21 g). ^1^H NMR (300 MHz, CDCl_3_) δ 1.24–1.70 (m, 31H, C–(CH_3_)_2_, H_3_C–(CH_2_)_11_), 3.2 (m, 1H, N^6^-H), 3.7 (d, *J* = 47.9 Hz, 3H, H-5′, H-4′, H-3′), 5.2 (d, *J* = 3.6 Hz, 2H, H-2′, H-1′), 5.78–5.83 (m, 1H, 5′OH), 7.90 (d, 1H, H-2), 8.39 (s, 1H, H-8). ^13^C NMR (75 MHz, CDCl_3_) δ 14.2–46.1 (m, C–(CH_3_)_2_, H_3_C–(CH_2_)_11_), 63.6 (C-5′), 81.9 (C-4′), 83.9 (C-3′), 86.7 (C-2′), 95.3 (C-1′), 114.3 (C-5), 121.4 (C–(CH_3_)_2_), 139.3 (C-8), 153.6 (C-2). HRMS [ESI] m/z: calcd. for C_25_H_42_N_5_O_4_ ([M+H]^+^) 476.3231, found: 476.3227.

#### 6.2.5. 2′,3′-isopropylidene-N^6^-(octadecyl)-adenosine (3d)

Compound **2** (200 mg, 0.613 mmol) was added to a microwave vial containing octadecylamine (248 mg, 0.92 mmol) and DIPEA (0.32 mL, 1.84 mmol) in DMSO (6 mL). The reaction condition, TLC and purification were similar as **3a**. Yield: 57% (0.20 g). HRMS [ESI] m/z: calcd. for C_31_H_54_N_5_O_4_ ([M+H]^+^) 560.4170, found: 560.4174.

#### 6.2.6. 2′,3′-isopropylidene-N^6^-(phenyl)-adenosine (3e)

Compound **2** (200 mg, 0.613 mmol) was added to a microwave vial containing aniline (0.083 mL, 0.92 mmol) and DIPEA (0.32 mL, 1.84 mmol) in DMSO (6 mL). The reaction condition, TLC and purification were similar as **3a**. Yield: 62% (0.15 g). ^1^H NMR (300 MHz, CDCl_3_) δ 1.3–1.6 (m, 6H, C–(C**H**_3_)_2_), 3.7–3.9 (m, 1H, N^6^-H), 3.9–4.0 (m, 2H, H-5′a, H-5′b), 4.5 (m, 1H, C-4′), 4.9–5.1 (m, 1H, C-3′), 5.2 (m, 1H, C-2′), 5.9 (d, *J* = 4.8 Hz, 1H, C-1′), 6.0 (d, *J* = 4.2 Hz, 1H, 5′O**H**), 6.4–7.6 (m, 5H, N^6^–C_6_**H**_5_), 8.4 (s, 1H, H-2), 8.7 (s, 1H, H-8). ^13^C NMR (75 MHz, CDCl_3_) δ 25.5 (C–(**C**H_3_)_2_), 27.9 (C–(**C**H_3_)_2_), 63.6 (C-5′), 81.8 (C-4′), 82.0 (C-3′), 83.5 (C-2′), 94.5 (C-1′), 114.3 (C-5), 114.7 (p-C-aniline), 121.0 (m-C-aniline), 122.0 (m-C-aniline), 124.3 (**C**-(CH_3_)_2_), 129.3 (o-C-aniline), 133.2 (o-C-aniline), 138.5 (N^6^-C-aniline), 140.5 (C-8), 145.0 (C-6), 148.3 (C-4), 152.3 (C-2). HRMS [ESI] m/z: calcd. for C_19_H_22_N_5_O_4_ ([M+H]^+^) 384.1661, found: 384.1664.

#### 6.2.7. 2′,3′-isopropylidene-5′-*O*-sulfamoyl-6-chloropurine riboside^20^ (3f)

Formic acid (1.73 mL, 46 mmol) was added dropwise to CSI (4.0 mL, 46 mmol) in an ice bath for 10 min. After several minutes, the formation of a white solid was observed. Afterwards, acetonitrile (20 mL) was added and the solution was stirred for four hours at room temperature. Following stirring, the solution was added to compound **2** (5 g, 15 mmol) in DMA (20 mL) and was reacted overnight at room temperature. Column chromatography was performed with a gradient of 15–25% acetone in hexane to obtain desired compound at 73% (4.53 g) yield.^1^H NMR (300 MHz, CDCl_3_) δ 1.36 (s, 3H, C–CH_3_), 1.60 (s, 3H, C–CH_3_), 4.29–4.44 (m, 2H, C’5–H_2_), 4.60 (m, 1H, H-4′), 5.06 (dd, *J* = 6.2, 2.7 Hz, 1H, H-3′), 5.34 (dd, *J* = 6.2, 2.6 Hz, 1H, H-2′), 6.25 (d, *J* = 2.6 Hz, 1H, H-1′), 6.31 (d, *J* = 5.1 Hz, 2H, SO_2_-NH_2_), 8.39 (s, 1H, H-2), 8.73 (s, 1H, H-8). ^13^C NMR (75 MHz, CDCl_3_) δ 25.4 (C–**C**H_3_), 27.2 (C-**C**H_3_), 69.4 (C-5′), 81.3 (C-4′), 84.4 (C-3′), 84.7 (C-2′), 91.5 (C-1′), 115.1 (**C**–(CH_3_)_2_), 132.0 (C-5), 144.4 (C-8), 151.2 (C-6), 152.4 (C-2). HRMS [ESI] m/z: calcd. for C_13_H_17_ClN_5_O_6_S ([M+H]^+^) 406.0582, found: 406.0574.

#### 6.2.8. 2′,3′-isopropylidene-5′-*O*-sulfamoyl-N^6^-methyl-adenosine (4a)

Formic acid (0.62 mL) was added dropwise to CSI (1.42 mL) in an RBF kept in an ice bath for 10 min. After a few minutes, the formation of a white solid was observed. Afterwards, acetonitrile (20 mL) was added, and the solution was stirred for four hours at room temperature. The obtained solution of sulfamoyl chloride is used in the next reaction as such. Sulfamoyl chloride solution (2.65 mL, 1.95 mmol) was added to a solution of compound **3a** (208 mg, 0.65 mmol) in DMA (10 mL) and was stirred overnight. TLC (50:50 acetone/hexane) was used to monitor the reaction. After overnight stirring, the solvent was evaporated under reduced pressure. Column chromatography was performed with a gradient of 40–50% acetone in hexane to obtain **4a**. Yield: 25% (0.065 g). ^1^H NMR (300 MHz, CDCl_3_) δ 1.3 (s, 3H, C–CH_3_), 1.6 (s, 3H, C–CH_3_), 3.0 (d, *J* = 7.0 Hz, 3H, NH-C**H**_3_), 4.4 (s, 2H, H-5′a, H-5′b), 4.5 (s, 1H, H-4′), 5.0 (d, *J* = 6.8 Hz, 1H, H-3′), 5.3 (d, *J* = 6.2 Hz, 1H, H-2′), 6.1 (d, *J* = 2.5 Hz, 1H, H-1′), 6.4 (s, 1H, N^6^-H), 7.9 (s, 1H, H-2), 8.3 (s, 1H, H-8). ^13^C NMR (75 MHz, CDCl_3_) δ 25.5 (C–**C**H_3_), 27.3 (C–**C**H_3_), 29.6 (N^6^–**C**H_3_), 69.6 (C-5′), 81.4 (C-4′), 84.4 (C-3′), 84.6 (C-2′), 90.9 (C-1′), 115.0 (C-5), 120.0 (**C**–(CH_3_)_2_), 139.1 (C-8), 153.6 (C-2), 155.5 (C-6). HRMS [ESI] m/z: calcd. for C_14_H_21_N_6_O_6_S ([M+H]^+^) 401.1238, found: 401.1236.

#### 6.2.9. 2′,3′-isopropylidene-5′-*O*-sulfamoyl-N^6^-octyl-adenosine (4b)

Sulfamoyl chloride solution (2.8 mL, 2.04 mmol) was added to compound **3b** (284 mg, 0.68 mmol, 0.33 equivalent) in DMA (10 mL). Similar reaction conditions and purification methods as described for the synthesis of compound **4a** were used to obtain **4b**. Yield: 97% (0.33 g). ^1^H NMR (300 MHz, CDCl_3_) δ 0.7–3.1 (m, 26H, C–(C**H**_3_)_2_, **H**_3_C–(C**H**_2_)_7_, NH–C**H**_3_), 3.5 (m, 2H, N^6^-H, H-5′a), 4.2–4.6 (m, 3H, SO_2_-NH_2_, H-5′b), 5.1 (dd, *J* = 6.3, 3.0 Hz, 1H, H-4′), 5.3 (dd, *J* = 6.5, 2.5 Hz, 1H, H-3′), 5.9–6.2 (m, 2H, H-2′, H-1′), 7.9 (d, *J* = 2.4 Hz, 1H, H-2), 8.3 (s, 1H, H-8). ^13^C NMR (75 MHz, CDCl_3_) δ 14.1–40.9 (C–(**C**H_3_)_2_, H_3_**C**–(**C**H_2_)_7_), 69.2 (C-5′), 81.2 (C-4′), 84.2 (C-3′), 84.4 (C-2′), 90.8 (C-1′), 114.8 (C-5), 119.9 (**C**-(CH_3_)_2_), 138.8 (C-8), 153.6 (C-2), 155.0 (C-6). HRMS [ESI] m/z: calcd. for C_21_H_35_N_6_O_6_S ([M+H]^+^) 499.2333, found: 499.2329.

#### 6.2.10. 2′,3′-isopropylidene-5′-*O*-sulfamoyl-N^6^-dodecyl-adenosine (4c)

Sulfamoyl chloride solution (2.4 mL, 1.76 mmol) was added to compound **3c** (210 mg, 0.44 mmol) in DMA (10 mL). Similar reaction conditions and purification methods as described for the synthesis of compound **4a** were used to obtain **4c**. Yield: 69% (0.17 g). ^1^H NMR (300 MHz, CDCl_3_) δ 0.7–3.6 (m, 33H, C–(C**H**_3_)_2_, **H**_3_C–(C**H**_2_)_11_, N^6^-H), 4.3–4.6 (m, 2H, SO_2_–NH_2_), 5.1 (m, 3H, H-5′a, H-5′b, H-4′), 5.4 (m, 2H, H-3′, H-2′), 5.9–6.1 (m, 1H, H-1′), 7.9 (s, 1H, H-2), 8.3 (s, 1H, H-8). HRMS [ESI] m/z: calcd. for C_25_H_43_N_6_O_6_S ([M+H]^+^) 555.2959, found: 555.2958.

#### 6.2.11. 2′,3′-isopropylidene-5′-*O*-sulfamoyl-N^6^-octadecyl-adenosine (4d)

Sulfamoyl chloride solution (1.95 mL, 1.44 mmol) was added to compound **3d** (200 mg, 0.36 mmol) in DMA (10 mL). Similar reaction conditions and purification methods as described for the synthesis of compound **4a** were used to obtain **4d**. Yield: 56% (0.13 g). ^1^H NMR (300 MHz, CDCl_3_) δ 0.9–1.3 (m, 44H, C–(C**H**_3_)_2_, **H**_3_C–(C**H**_2_)_17_, N^6^-H), 4.3–4.6 (m, 3H, SO_2_–NH_2_, H-5′a), 5.1 (dd, *J* = 6.4, 3.0 Hz, 1H, H-5′b), 5.3 (dd, *J* = 6.3, 2.6 Hz, 1H, H-4′), 5.8–6.2 (m, 3H, H-3′, H-2′, H-1′), 7.9 (s, 1H, H-2), 8.3 (s, 1H, H-8). ^13^C NMR (75 MHz, CDCl_3_) δ 14.4–32.2 (C–(**C**H_3_)_2_, H_3_**C**–(**C**H_2_)_17_), 69.7 (C-5′), 81.3 (C-4′), 84.2 (C-3′), 84.6 (C-2′), 91.1 (C-1′), 115.1 (**C**-(CH_3_)_2_), 139.2 (C-8), 153.8 (C-2). HRMS [ESI] m/z: calcd. for C_25_H_43_N_6_O_6_S ([M+H]^+^) 555.2959, found: 555.2958.

#### 6.2.12. 2′,3′-isopropylidene-5′-*O*-sulfamoyl-N^6^-phenyl-adenosine (4e)

Sulfamoyl chloride solution (2 mL, 1.47 mmol) was added to compound **3e** (188 mg, 0.68 mmol) in DMA (10 mL). Reaction conditions and purification methods were as described for **4a** to obtain **4e**. Yield: 76% (0.24 g).^1^H NMR (300 MHz, CDCl_3_) δ 1.4–1.6 (m, 6H, C-(C**H**_3_)_2_), 2.1–3.0 (m, 3H, N^6^-H, SO_2_NH_2_), 4.4 (m, 2H, H-5′a, H-5′b), 4.5–4.7 (m, 1H, H-4′), 5.1 (m, 1H, H-3′), 5.3–5.4 (m, 1H, H-2′), 6.2 (m, 1H, H-1′), 6.5 (s, 1H, p-CH-aniline), 7.4 (m, 2H, 2x m-CH-aniline), 7.7–7.8 (m, 1H, o-H-aniline), 8.1 (d, *J* = 3.2 Hz, 1H), 8.4 (d, *J* = 18.6 Hz, 1H, H-2), 8.7 (d, *J* = 2.1 Hz, 1H, H-8). ^13^C NMR (75 MHz, CDCl_3_) δ 25.5 (C–(**C**H_3_)_2_), 27.4 (C–(**C**H_3_)_2_), 69.6 (C-5′), 81.5 (C-4′), 84.6 (C-3′), 84.8 (C-2′), 91.2 (C-1′), 115.0 (C-5), 115.2 (p-C-aniline), 120.9 (m-C-aniline), 124.0 (m-C-aniline), 129.2 (o-C-aniline), 138.8 (N^6^-C-aniline), 139.9 (o-C-aniline), 144.5 (C-8), 149.2 (C-6), 151.3 (C-4), 152.5 (C-2). HRMS [ESI] *m/z*: calcd. for C_19_H_22_N_6_O_6_S ([M+H]^+^) 463.1394, found: 463.1385.

#### 6.2.13. 2′, 3′-isopropylidene-5′-*O*-sulfamoyl-6-*O*-methyl-purine riboside (4f)

A 20% solution of sodium methoxide in methanol (0.27 mL, 0.98 mmol) was added to a cooled solution of **3f** in dry methanol. The reaction was performed at 0 °C. After two hours, a few drops of glacial acetic acid were added to quench the reaction. The reaction mixture was evaporated under reduced pressure. Column chromatography was performed with a gradient of 40% acetone in hexane to yield **4f**. Yield: 39% (0.15 g). ^1^H NMR (300 MHz, CDCl_3_) δ 1.3 (s, 3H, C–CH_3_), 1.4 (s, 3H, C–CH_3_), 4.1 (s, 3H, O–CH_3_), 4.3–4.5 (m, 2H, H-5′a, H-5′b), 4.6 (s, 1H, H-4′), 5.1 (dd, *J* = 6.4, 2.8 Hz, 1H, H-3′), 5.4 (dd, *J* = 6.2, 2.4 Hz, 1H, H-2′), 6.2 (d, *J* = 2.4 Hz, 1H, H-1′), 8.1 (s, 1H, H-2), 8.5 (s, 1H, H-8). ^13^C NMR (75 MHz, CDCl_3_) δ 25.5 (C-CH_3_), 27.4 (C–CH_3_), 54.6 (O-CH_3_), 69.8 (C-5′), 81.5 (C-4′), 84.5 (C-3′), 84.8 (C-2′), 91.6 (C-1′), 115.2 (C-5), 122.1 (C– (CH_3_)_2_), 141.5 (C-8), 151.4 (C-2), 152.8 (C-6).

#### 6.2.14. 2′,3′-isopropylidene-5′-*O*-(*N*-(N^α^-Boc-L-isoleucyl))-sulfamoyl-N^6^-methyl-adenosine (5a)

Compound **4a** (100 mg, 0.27 mmol) was added to a solution of Boc-Ile-Osu (133 mg, 0.405 mmol) and DBU (0.06 mL, 0.405 mmol) in DMF (12 mL). The solution was stirred overnight at room temperature. TLC was developed with a methanol/ethyl acetate mixture (10:90) to monitor the reaction. After stirring overnight, the solvent was evaporated under reduced pressure. Column chromatography was performed with a mixture of 5% methanol in ethyl acetate to obtain compound **5a**. Yield: 85% (0.14 g). ^1^H NMR (300 MHz, CDCl_3_) δ 0.8–3.4 (m, 30H, Boc-Ile protons, C-(CH_3_)_2_, NH-CH_3_), 3.9 (s, 1H, N^6^-H), 4.1 (m, 1H, H-5′a), 4.2 (m, 3H, H-5′b, H-4′, H-3′), 4.5 (s, 1H, H-2′), 6.2 (d, *J* = 2.5 Hz, 1H, H-1′), 8.3 (d, *J* = 12.1 Hz, 2H, H-2, H-8). ^13^C NMR (75 MHz, CDCl_3_) δ 11.6–62.2 (C-(CH_3_)_2_), Boc-Ile carbons), 69.1 (C-5′), 81.5 (C-4′), 85.1 (C-3′), 90.7 (C-1′), 114.3 (C-5), 119.5 (C–(CH_3_)_2_), 139.6 (C-8), 148.6 (C-4), 153.7 (C-6), 155.3 (-**C**(O)-tBu). HRMS [ESI] m/z: calcd. for C_25_H_39_N_7_O_9_S ([M-H]^-^) 612.2457, found: 612.2462.

#### 6.2.15. 2′,3′-isopropylidene-5′-*O*-(*N*-(N^α^-Boc-L-isoleucyl))-sulfamoyl-N^6^-octyl-adenosine (5b)

Compound **4b** (350 mg, 0.763 mmol) was added to a solution of Boc-Ile-Osu (376 mg, 1.145 mmol) and DBU (0.171 mL, 1.145 mmol) in DMF (12 mL). Reaction conditions and purification were similar as described for the synthesis of **5a**, except that TLC and column chromatography were performed with a mixture of 30% acetone in hexane, to obtain **5b**. Yield: 87% (0.47 g). ^1^H NMR (300 MHz, CDCl_3_) δ 0.8 - 3.6 (m, 44H, Boc-Ile protons, C–(CH_3_)_2_, NH(CH_2_)_7_CH_3_), 3.7 (d, *J* = 11.8 Hz, 1H, N^6^–H), 4.3 (m, 2H, H-5′a, H-5′b), 4.5 (d, *J* = 5.8 Hz, 1H, H-4′), 4.5 (s, 1H, H-3′), 4.9 (s, 1H, H-2′), 6.2 (s, 1H, H-1′), 8.2 (d, *J* = 15.9 Hz, 2H, SO_2_NH, N^α^-H), 8.4 (s, 1H, H-2), 8.6 (s, 1H, H-8). HRMS [ESI] m/z: calcd. for C_32_H_53_N_7_O_9_S ([M-H]^-^) 710.3524, found: 710.3397.

#### 6.2.16. 2′,3′-isopropylidene-5′-*O*-(*N*-(N^α^-Boc-L-isoleucyl))-sulfamoyl-N^6^-dodecyl-adenosine (5c)

Compound **4c** (200 mg, 0.36 mmol) was added to a solution of Boc-Ile-OSu (177 mg, 0.54 mmol) and DBU (0.08 mL, 0.54 mmol) in DMF (12 mL). Reaction conditions and purification were similar as described for the synthesis of **5a**, except that TLC and column chromatography were carried out with a mixture of 35% acetone in hexane, to obtain **5c**. Yield: 51% (0.14 g). ^1^H NMR (300 MHz, CDCl_3_) δ 1.4 –4.0 (m, 52H, Boc-Ile protons, C–(C**H**_3_)_2_, NH(C**H**_2_)_11_C**H**_3_), 4.2 (s, 1H, N^6^–H), 4.6 (s, 1H, H-5′a), 4.9 (s, 1H, H-5′b), 5.2 (s, 1H, H-4′), 5.7 (s, 1H, H-3′), 5.8 (s, 1H, H-2′), 8.9 (s, 1H, H-8). ^13^C NMR (75 MHz, CDCl_3_) δ 11.8–44.7 (C-(**C**H_3_)_2_), Boc-Ile carbons), 68.7 (C-5′), 80.5 (C-4′), 85.1 (C-3′), 85.9 (C-2′), 92.5 (C-1′), 114.3 (C-5), 153.7 (C-6), 155.0 (-**C**(O)-tBu). HRMS [ESI] *m/z*: calcd. for C_36_H_61_N_7_O_9_S ([M-H]^-^) 766.41784, observed 766.4161.

#### 6.2.17. 2′,3′-isopropylidene-5′-*O*-(*N*-(N^α^-Boc-L-isoleucyl))-sulfamoyl-N^6^-octadecyl-adenosine (5d)

Compound **4d** (150 mg, 0.234 mmol) was added to a solution of Boc-Ile-Osu (115 mg, 0.35 mmol) and DBU (0.05 mL, 0.35 mmol) in DMF (12 mL). Reaction conditions and were similar as described for **5a**, except that TLC and column chromatography were performed with a mixture of 35% acetone in hexane, to obtain **5d**. Yield: 43% (0.12 g). ^1^H NMR (300 MHz, CDCl_3_) δ 0.8 - 3.6 (m, 52H, Boc-H, C-(C**H**_3_)_2_, NH(C**H**_2_)_17_C**H**_3_), 4.0 (s, 1H, N^6^-H), 4.3 (s, 2H, H-5′a, H-5′b), 4.6 (s, 1H, H-4′), 5.0–5.1 (m, 1H, H-3′), 5.2 (s, 1H, H-2′), 5.4 (s, 1H, H-1′), 6.3 (s, 1H, N^α^-H), 6.6 (s, 1H, SO_2_NH), 8.4 (s, 1H, H-2). HRMS [ESI] *m/z*: calcd. for C_42_H_73_N_7_O_9_S ([M-H]^-^) 850.5117, found: 850.5115.

#### 6.2.18. 2′,3′-isopropylidene-5′-*O*-(*N*-(N^α^-Boc-L-isoleucyl))-sulfamoyl-N^6^-phenyl-adenosine (5e)

Compound **4e** (250 mg, 0.60 mmol) was added to a solution of Boc-Ile-Osu (295 mg, 0.90 mmol) and DBU (0.134 mL, 0.90 mmol) in DMF (12 mL). Reaction conditions and purification were similar as described for **5a**, except that TLC and column chromatography were carried out with a mixture of 35% acetone in hexane to obtain **5e**. Yield: 49% (0.20 g). ^1^H NMR (300 MHz, CDCl_3_) δ 1.4–2.1 (m, 22H, Boc-Ile-protons, C–(C**H**_3_)_2_), 4.5 (s, 1H, H-C^α^), 4.6 (s, 1H, H-5′a), 4.9 (s, 1H, H-5′b), 5.2 (s, 1H, H-4′), 5.5 (d, *J* = 14.3 Hz, 1H, H-3′), 5.7 (s, 1H, H-2′), 6.9 (d, *J* = 7.7 Hz, 1H, H-1′), 7.7 (m, 1H, p-CH-aniline), 7.8 –8.0 (m, 2H, m-CH-aniline), 8.3 (d, *J* = 8.0 Hz, 2H, o-CH-aniline), 9.0 (s, 1H, H-2), 9.1 (s, 1H, H-8). ^13^C NMR (75 MHz, CDCl_3_) δ 11.8–62.2 (C–(**C**H_3_)_2_), Boc-Ile carbons), 69.2 (C-5′), 80.5 (C-4′), 85.3 (C-3′), 85.8 (C-2′), 91.3 (C-1′), 114.5 (C-5), 120.3 (p-C-aniline), 121.2 (m-C-aniline), 121.8 (m-C-aniline), 123.2 (**C**–(CH_3_)_2_), 123.8 (o-C-aniline), 126.9 (o-C-aniline), 139.0 (N^6^-C-aniline), 139.5 (C-8), 149.5 (C-6), 152.3 (C-4), 152.8 (C-2), 156.7 (-**C**(O)-tBu). HRMS [ESI] *m/z*: calcd. for C_30_H_41_N_7_O_9_S ([M-H]^-^) 674.2613, found: 674.2592.

#### 6.2.19. Synthesis of 2′, 3′-isopropylidene-5′-*O*-sulfamoyl-(*N*-(N^α^-Boc-L-isoleucyl)-6-*O*-methyl-purine riboside (5f)

Compound **4f** (150 mg, 0.37 mmol) was added to a solution of Boc-Ile-Osu (184 mg, 0.54 mmol) and DBU (0.08 mL, 0.54 mmol) in DMF (12 mL). The reaction was stirred overnight at room temperature. The solvent was evaporated under reduced pressure. Column chromatography of the obtained residue was carried out with a gradient of 10–20% acetone in hexane to yield **5f**. Yield: 92% (0.21 g).

#### 6.2.20. Synthesis of 5′-*O*-(*N*-L-isoleucyl)-sulfamoyl-N^6^-methyl-adenosine (6a)

Compound **5a** (120 mg, 0.20 mmol) was dissolved in a mixture of TFA, water and DCM (50:25:25 *v/v/v*) and stirred for 3 h at room temperature. TLC was performed in a methanol/DCM mixture (10:90 *v/v*) to check the reaction progression. After 3h, the mixture was evaporated at reduced pressure at 25 °C. RP-HPLC was performed with a C18 column with gradient elution using water/acetonitrile as the mobile phase to purify and obtain **6a**. Yield: 37% (0.035 g). ^1^H NMR (300 MHz, D_2_O) δ 0.90–1.02 (m, 6H, Ile-δ-CH_3_, Ile-γ-CH_3_), 1.20-1.26 (m, 1H, Ile-γ-CH_2_ Ha), 1.56–1.61 (m, 1H, Ile-γ-CH_2_ Hb), 1.95-1.99 (m, 1H, Ile-β-CH), 3.11 (bs, 3H, N–CH_3_), 3.56 (d, *J* = 4.1 Hz, 1H, Ile-α-CH), 4.29–4.39 (m, 4H, H-5′, H-5″, H-4′, H-3′), 4.62 (t, 1H, *J* = 5.0 Hz, H-2′), 6.06 (s, 1H, *J* = 5.3 Hz, H-1′), 8.24 (s, 1H, H-2), 8.45 (s, 1H, H-8). ^13^C NMR (75 MHz, D_2_O) δ 12.1 (Ile- δ -CH_3_), 15.5 (Ile-γ-CH_3_), 25.7 (Ile- δ-CH_2_), 38.2 (Ile-β-CH), 61.5 (Ile-α-CH), 69.0 (C-5′), 72.0 (C-4′), 76.2 (C-3′), 84.2 (C-2′), 89.4 (C-1′), 120.6 (C-5), 140.6 (C-8), 153.9 (C-2), 156.7 (C-6), 174.9 (-**C**(C=O, Ile). HRMS [ESI] m/z: calcd. for C_17_H_27_N_7_O_7_S ([M-H]^-^) 472.1620, found: 472.1633.

#### 6.2.21. Synthesis of 5′-*O*-(*N*-L-isoleucyl)-sulfamoyl-N^6^-octyl-adenosine (6b)

Compound **5b** (300 mg, 0.42 mmol) was treated like compound **5a** regarding reaction condition and purification to obtain **6b**. Yield: 21% (0.051 g). ^1^H NMR (300 MHz, MeOD) δ 0.90–0.95 (m, 6H, CH_3_-Octyl chain and Ile-δ-CH_3_), 1.03 (d, *J* = 7.0 Hz, 3H, Ile-γ-CH_3_), 1.32-1.46 (m, 11H, Octyl chain and Ile-γ-CH_2_ Ha), 1.58–1.62 (m, 1H, Ile-γ-CH_2_ Hb), 1.70–172 (m, 2H CH_2_-Octyl chain), 1.97–2.01 (m, 1H, Ile-β-CH), 3.56 (d, *J* = 3.7 Hz, 1H, Ile-α-CH), 3.58–3.62 (m, 2H, NH-CH_2_ octyl chain), 4.31–4.42 (m, 4H, H-5′a, H-5′b, H-4′, H-3′), 4.64 (t, *J* = 5.1 Hz, 1H, H-2′), 6.10 (d, *J* = 5.4 Hz, 1H, H-1′), 8.25 (s, 1H, H-2), 8.48 (s, 1H, H-8). ^13^C NMR (75 MHz, D_2_O) δ 12.1 (Ile- δ -CH_3_), 14.4 (C8 alkyl chain), 15.5 (Ile-γ-CH_3_), 23.7(C8 alkyl chain), 25.6 (Ile- δ-CH_2_), 28.0–33.0 (C8 alkyl chain), 38.2 (Ile-β-CH), 41.7 (C8 alkyl chain), 61.5 (Ile-α-CH), 69.1 (C-5′), 72.0 (C-4′), 76.2 (C-3′), 84.3 (C-2′), 89.3 (C-1′), 120.4 (C-5), 140.4 (C-8), 149.7 (C-6), 154.0 (C-4), 156.1 (C-2), 175.0 (C=O, Ile). HRMS [ESI] *m/z*: calcd. for C_24_H_41_N_7_O_7_S ([M+H]^+^) 572.2861, found 572.2864.

#### 6.2.22. 5′-*O*-(*N*-L-isoleucyl)-sulfamoyl-N^6^-dodecyl-adenosine (6c)

Compound **5c** (90 mg, 0.12 mmol) was treated in the same way as compound **5a** regarding reaction condition and purification to obtain **6c**. Yield: 11% (0.008 g). ^1^H NMR (300 MHz, MeOD) δ 0.87–0.90 (m, 6H, CH_3_-dodecyl chain and Ile-δ-CH_3_), 1.04 (d, *J* = 7.0 Hz, 3H,), 1.28–1.30 (m, 31H, dodecyl chain, Ile-γ-CH_3_), 1.68–1.73 (m, 4H, Ile-β-CH, dodecyl chain), 3.37–3.39 (m, 2H, NH-CH_2_ dodecyl chain), 3.71–3.77 (m, 3H, Ile-α-CH), 4.51–4.66 (m, 5H, H-5′a, H-5′b, H-3′, H-4′), 4.89–4.90 (s, 1H, H-2′), 6.55 (s, 1H, H-1′), 8.31 (d, *J* = 4.8 Hz, 1H, H-2), 8.56 (s, 1H, H-8). HRMS [ESI] m/z: calcd. for C_28_H_49_N_7_O_7_S ([M-H]^-^): 626.3341, found: 626.3338.

#### 6.2.23. 5′-*O*-(*N*-L-isoleucyl)-sulfamoyl-N^6^-octadecyl-adenosine (6d)

Compound **5d** (80 mg, 0.09 mmol) was treated in the same way as compound **5a** to obtain **6d**. Yield: 19% (0.012 g). ^1^H NMR (300 MHz, MeOD) δ 0.89-0.98 (m, 8H, CH_3_-octadecyl chain and Ile-δ-CH_3_), 1.04 (d, *J* = 7.0 Hz, 3H, Ile-γ-CH_3_), 1.30 (s, 31H, octadecyl chain), 1.55–1.61 (m, 1H, Ile-γ-CH_2_ Hb), 1.69-171 (m, 2H CH_2_-Octadecyl chain), 1.97-2.01 (m, 1H, Ile-β-CH), 3.56-362 (m, 3H, Ile-α-CH and NH-CH_2_ octadecyl chain), 4.31–4.41 (m, 5H, H-5′a, H-5′b, H-3′, H-2′, H-4′), 6.09 (d, *J* = 5.4 Hz, H-1′), 8.25 (d, *J* = 4.8 Hz, 1H, H-2), 8.48 (s, 1H, H-8). HRMS [ESI] m/z: calcd. for C_34_H_61_N_7_O_7_S ([M+H]^+^) 712.4426, found: 712.4470.

#### 6.2.24. Synthesis of 5′-*O*-(*N*-L-isoleucyl)-sulfamoyl-N^6^-phenyl-adenosine (6e)

Compound **5e** (150 mg, 0.22 mmol) was treated as for **5a** to obtain **6e**. Yield: 18% (0.021 g). ^1^H NMR (300 MHz, MeOD) δ 0.90–0.95 (m, 3H, Ile-δ-CH_3_,), 1.03 (d, *J* = 6.9 Hz, 3H, Ile-γ-CH_3_), 1.20–1.33 (m, 1H, Ile-γ-CH_2_ Ha), 1.56–1.60 (m, 1H, Ile-γ-CH_2_ Hb), 1.97–2.01 (m, 1H, Ile-β-CH), 3.61 (d, *J* = 3.9 Hz, 1H, Ile-α-CH), 4.37–4.45 (m, 5H, H-5′, H-5″, H-4′, H-3′, H-2′), 6.14 (d, *J* = 5.1 Hz, 1H, H-1′), 7.14 (t, *J* = 7.1 Hz, 1H, p-CH-aniline), 7.39 (t, *J* = 7.5 Hz, 2H, m-CH-aniline), 7.76 (d, *J* = 8.1 Hz, 2H, o-CH-aniline), 8.38 (s, 1H, H-2), 8.57 (s, 1H, H-8). ^13^C NMR (75 MHz, D_2_O) δ 10.5 (Ile- δ -CH_3_), 13.9 (Ile-γ-CH_3_), 23.9 (Ile- δ-CH_2_), 36.5 (Ile-β-CH), 59.8 (Ile-α-CH), 67.5 (C-5′), 70.2 (C-4′), 74.4 (C-3′), 82.5 (C-2′), 87.7 (C-1′), 119.4 (C-5), 120.7 (o-CH-aniline), 123.4 (p-CH-aniline), 128.3 (m-CH-aniline), 138.3 (NH-CH-aniline), 139.8 (C-8), 142.6 (C-6), 149.0 (C-4), 152.0 (C-2), 173.7 (C=O, Ile). HRMS [ESI] m/z: calcd. for C_22_H_29_N_7_O_7_S ([M-H]^-^): 534.1776, found 534.1777.

#### 6.2.25. Synthesis of 5′-*O*-(*N*-L-isoleucyl)-sulfamoyl-6-*O*-methyl-purine riboside (6f)

Compound **5f** (100 mg, 0.16 mmol) was added to a fresh solution of TFA/water/DCM (50:25:25 v/v), and the reaction was stirred for 3 h at room temperature. TLC, with an acetone/hexane gradient, was used to monitor the reaction. The reaction mixture was evaporated under reduced pressure at 25 °C. The product was purified with RP-HPLC using a C18 column and gradient elution, with water/acetonitrile as the mobile phase, to obtain **6f**. Yield: 55% (0.042 g). ^1^H NMR (300 MHz, MeOD) δ 0.91–1.03 (m, 6H, Ile-δ-CH_3_, Ile-γ-CH_3_), 1.28–1.29 (m, 1H, Ile-γ-CH_2_ Ha), 1.54–1.56 (m, 1H, Ile-γ-CH_2_ Hb), 1.95–1.97 (m, 1H, Ile-β-CH), 3.58 (d, *J* = 4.0 Hz, 1H, Ile-α-CH), 4.18 (s, 3H, O-CH_3_), 4.32–4.43 (m, 4H, H-5′a, H-5′b, H-4′, H-3′), 4.66–4.69 (m, 1H, H-2′), 6.17 (d, *J* = 5.2 Hz, 1H, H-1′), 8.53 (s, 1H, H-2), 8.68 (s, 1H, H-8). ^13^C NMR (75 MHz, D_2_O) δ 12.1 (Ile- δ -CH_3_), 15.4 (Ile-γ-CH_3_), 25.7 (Ile- δ-CH_2_), 38.2 (Ile-β-CH), 54.8 (O-Me), 61.4 (Ile-α-CH), 69.0 (C-5′), 72.0 (C-4′), 76.2 (C-3′), 84.4 (C-2′), 89.8 (C-1′), 143.5 (C-2), 153.5 (C-2). HRMS [ESI] m/z: calcd. for C_17_H_26_N_6_O_8_S ([M-H]^-^) 473.1460, found: 473.1471.

The synthesis of compound 2′,3′,5′-tri-*O*-TBDMS-adenosine (**8**), 2′,3′-di-*O*-TBDMS adenosine (**9**), and 2′,3′-di-*O*-TBDMS-5′-*O*-sulfamoyl adenosine (**10**) was performed by following reported procedure^20^ and obtained at 88, 84 and 97% yields respectively.

#### 6.2.26. 5′-*O*-[*N*-(*N*-Boc)leucyl]sulfamoyl adenosine (11)

Compound **10** (300 mg, 0.52 mmol), Boc-Leu-OSu (1.2 equivalent, 205.85 mg, 0.63 mmol), DBU (1 equivalent, 0.08 mL, 0.52 mmol) were added together using DMF (10 mL) as solvent. The reaction, which after a short period of time turned pink, was let to react at room temperature and overnight. A small sample was work-upped with EtOAc and 10% KHSO_4_ for TLC analysis which was later developed with 1% MeOH in EtOAc and sprayed with ammonium molybdate (R_f_ = 0.73). The solvents were evaporated after completion of the reaction and the obtained wine-red dense liquid was partitioned between water and EtOAc. A small amount of 10% KHSO_4_ was added in the first wash to assure that the pH from the aqueous layer had pH 5-6. The organic layers were collected, dried over MgSO_4_, filtered and, lastly, the solvent was evaporated. The residue was purified using silica gel column chromatography with elution at 1.5% MeOH:EtOAc. The UV-active fractions were collected and further dried. Yield: 77%. ^1^H NMR (300 MHz, MeOD) δ -0.42 (3H, s, CH_3_-Si), -0.18 (3H, m, CH_3_-Si), 0.11–0.12 (6H, m, CH_3_-Si), 0.66 (9H, s, ^t^Bu CH_3_), 0.84–0.86 (6H, m, Leu-δ’-CH_3_, Leu-δ″), 0.92 (9H, s, ^t^Bu CH_3_), 1.36 (10H, s, Boc-^t^Bu CH_3_ Leu-γ-CH), 1.47–1.63 (2H, m, Leu-ß-CH_2_), 3.77-3.78 (1H, m, Leu-α-CH), 4.02–4.14 (3H, m, H_5_″, H_5_’, H_3_’), 4.32 (1H, d, *J* = 7.4 Hz, H_2_’), 4.89–4.93 (1H, m, H_4_’), 5.95 (1H, d, *J* = 7.4 Hz, H_1_’), 7.26 (2H, bs, NH_2_), 8.13 (1H, s, H_8_), 8.47 (1H, s, H_2_). ^13^C NMR (300 MHz, MeOD), δ -5.6 (CH_3_-Si), -5.7 (CH_3_-Si), -4.6 (CH_3_-Si), 17.5 (^t^Bu C(CH_3_)_3_), 17.9 (^t^Bu C(CH_3_)_3_), 22.1 (Leu-δ’-CH_3_), 23.4 (Leu-δ″-CH_3_), 24.5 (Leu-γ-CH), 25.6 (^t^Bu C(CH_3_)_3_), 25.9 (^t^Bu C(CH_3_)_3_), 28.4 (Boc-^t^Bu CH_3_), 43.0 (Leu-ß-CH_2_), 54.9 (Leu-α-CH), 66.9 (C-5′), 73.4 (C-3′), 74.7 (C-2′), 77.5 (C-4′), 84.1 (Boc-tertiary C), 86.1 (C-1′), 119.0 (C-5), 139.7 (C-8)150.0 (C-4), 152.8 (C-2), 155.1 (Boc- carbonyl C), 156.1(C-6), 177.0 (CONH). HRMS [ESI] m/z: calcd. for C_33_H_60_N_7_O_9_S_1_Si_2_ ([M-H]^-^) 786.3717, found: 786.3720.

#### 6.2.27. 2′,3′-di-*O*-TBDMS-5′-*O*-(*N*-leucyl)sulfamoyl adenosine (12)

Selective Boc deprotection was performed by adding a 10 mL solution of TFA:DCM:water (2:1:1) to the Boc and TBDMS protected compound **9** (273 mg). The reaction was first started at 0 °C and then held at room temperature for 2 h. For TLC (20% MeOH in EtOAc plus a few drops of TEA was used as mobile phase; R_f_ = 0.3), the sample was first evaporated in high-vacuum, then the dried compound was dissolved and evaporated two times after dissolving with EtOH to assure that there was no TFA left, as it could interfere with TLC analysis. Upon completion of the reaction, TFA and DCM were evaporated, and the resulting residue was purified by silica gel column chromatography. ^1^H NMR (300 MHz, MeOD) δ -0.36 (3H, s, CH_3_-Si), 0.06 (3H, m, CH_3_-Si), 0.11–0.14 (6H, m, CH_3_-Si), 0.69 (9H, s, ^t^Bu CH_3_), 0.88–0.93 (15H, m, ^t^Bu CH_3,_ Leu-δ’-CH_3_, Leu-δ″), 1.47–1.53 (1H, m, Leu-γ-CH), 1.69–1.77 (2H, m, Leu-ß-CH_2_), 3.48–3.53 (1H, m, Leu-α-CH), 4.24–4.39 (4H, m, H_5_″, H_5_’, H_3_’, H_2_’), 4.76–4.80 (1H, m, H_4_’), 6.06 (1H, d, *J* = 7.1 Hz, H_1_’), 8.14 (1H, s, H_8_), 8.53 (1H, s, H_2_). ^13^C NMR (300 MHz, MeOD), δ -5.2 (CH_3_–Si), -4.3 (CH_3_–Si), -4.2 (CH_3_–Si), 18.7 (^t^Bu C(CH_3_)_3_), 18.9 (^t^Bu C(CH_3_)_3_), 22.3 (Leu-δ’-CH_3_), 23.4 (Leu-δ″-CH_3_), 25.8 (Leu-γ-CH), 26.2 (^t^Bu C(CH_3_)_3_), 26.4 (^t^Bu C(CH_3_)_3_), 43.3 (Leu-ß-CH_2_), 55.9 (Leu-α-CH), 69.1 (C-5′), 74.6 (C-3′), 77.6 (C-2′), 85.9 (C-4′), 88.5 (C-1′), 120.1 (C-5), 141.6 (C-8), 151.0 (C-4), 153.9 (C-2), 157.3(C-6), 178.6 (CONH). HRMS [ESI] m/z: calcd. for C_28_H_52_N_7_O_7_S_1_Si_2_ ([M-H]^-^) 686.3193, found 686.3195.

#### 6.2.28. 2′,3′-di-*O*-TBDMS-5′-*O*-[N^α^-(p-nitrobenzyloxycarbonyl)leucyl]sulfamoyl adenosine (13)

For coupling of the leucyladenosine derivative and the promoiety, the dried compound **12** (116 mg, 0.17 mmol) was mixed with 4-nitrobenzyl chloroformate (1.2 equivalent, 43.62 mg, 0.204 mmol) and DIPEA (3 equivalent, 0.09 mL, 0.51 mmol) and dissolved in DFM. The reaction was left at room temperature overnight. A predeveloped TLC was eluted at 5% MeOH in DCM in the presence of a few drops of TEA (R_f_ = 0.42). The solvent was evaporated in order to obtain a residue and silica gel column chromatography was used to purify the compound. Yield: 62%. ^1^H NMR (300 MHz, MeOD) δ -0.21–0.20 (3H, s, CH_3_–Si), 0.11–0.14 (3H, m, CH_3_–Si), 0.29–0.31 (6H, m, CH_3_–Si), 0.86–0.92 (9H, m, Leu-δ’-CH_3_, Leu-δ″-CH_3_, Leu-γ-CH, Leu-ß-CH_2_), 1.01–1.11 (18H, m, 2·^t^Bu CH_3_), 4.43–4.57 (6H, m, H_5_″, H_5_’, H_4_’, H_3_’, H_2_’, Leu-α-CH), 5.34 (2H, s, Bn CH_2_), 6.21–6.26 (1H, m, H_1_’), 7.69-7.71 (2H, d, o-2CH), 8.31-8.33 (3H, d, H_8_, m-2H), 8.70 (1H, m, H_2_). ^13^C NMR (300 MHz, MeOD), δ -5.2 (CH_3_–Si), -4.3 (CH_3_–Si), -4.2 (CH_3_–Si), 22.1 (Leu-δ’-CH_3_), 23.4 (Leu-δ″-CH_3_), 23.8 (Leu-γ-CH), 26.2 (^t^Bu C(CH_3_)_3_), 26.4 (^t^Bu C(CH_3_)_3_), 42.5 (Leu-ß-CH_2_), 55.2 (Leu-α-CH), 61.1 (C-5′), 66.1 (Bn), 74.6 (C-3′), 77.4 (C-2′), 85.9 (C-4′), 88.3 (C-1′), 120.1 (C-5), 124.5 (o-CH), 130.0 (m-CH),141.5 (C), 145.5 (p-CH), 148.8 (C-4), 151.0 (C-2), 153.9 (C-6) 157.3 (COO-Bn), 177.95 (CONH). HRMS [ESI] m/z: calcd. for C_36_H_58_N_8_O_11_S_1_Si_2_ ([M-H]^-^) 865.3411, found: 865.3431.

#### 6.2.29. 5′-*O*-[N^α^-(p-nitrobenzyloxycarbonyl)leucyl]sulfamoyl adenosine (14)

First, compound **13** (160 mg, 0.19 mmol) was dissolved in THF (5 mL) and Et_3_N·3HF (2 mL) was added to start the reaction at room temperature for overnight. The reaction progress was checked with TLC which was predeveloped in 10% MeOH: DCM in the presence of TEA (R_f_ = 0.48). The reaction mixture was purified using silica gel chromatography. Fractions corresponding to compound **14** were collected, evaporated and further purified using reverse phase HPLC. ^1^H NMR (300 MHz, MeOD) δ 0.89–0.95 (6H, m, Leu-δ’-CH_3_, Leu-δ″-CH_3_) 1.70–1.78 (1H, m, Leu-γ-CH), 1.90–1.92 (2H, m, Leu-ß-CH_2_), 4.19–4.38 (5H, m, H_5_″, H_5_’, H_4_’, H_3_’, Leu-α-CH), 4.63–4.66 (1H, m, H_2_’), 5.18–5.22 (2H, m, Bn-CH_2_), 6.06–6.08 (1H, d, H_1_’, *J* = 5.73), 7.55–7.57 (2H, d, o-2CH, *J* = 8.70), 8.17–8.20 (3H, m, H_8_, m-2CH), 8.49 (1H, s, H_2_). ^13^C NMR (300 MHz, MeOD), 20.3 (Leu-δ’-CH_3_), 21.6 (Leu-δ″-CH_3_), 24.3 (Leu-γ-CH), 40.7 (Leu-ß-CH_2_), 59.4 (Leu-α-CH), 64.5 (C-5′), 67.7 (Bn), 70.5 (C-3′), 74.3 (C-2′), 82.7 (C-4′), 87.4 (C-1′), 118.4 (C-5), 122.8 (m-CH), 127.4 (o-CH), 139.4 (ipso-CH), 144.3 (C-8), 147.1 (C-4), 149.2 (p-CH), 152.2 (C-2), 155.5 (C-6), 171.2 (CO–**N^α^**H), 178.02 (CONH). HRMS [ESI] m/z: calcd. for C_24_H_30_N_8_O_11_S_1_ ([M-H]^-^) 637.1682 found: 637.1689.

#### 6.2.30. p-Acetoxybenzylchloroformate (16)

Take triphosgene (134mg, 0.45 mmol) in a round bottom flask and dry using high vacuum; then at 0 °C add DIPEA (0.16 mL, 0.9 mmol, 2 equiv.) and 5 mL THF. In another round bottom flask weigh 75 mg of p-acetoxybenzylalcohol and add 5 mL THF, then add this solution to triphosgene at 0 °C. Let the reaction continue at 0 °C for four hours. Use the reaction mixture without purification for the next reaction. The formation of the compound was confirmed by NMR. ^1^H NMR (300 MHz, CDCl_3_) δ 2.29 (s, 3H, acetyl CH_3_), 5.09(s, 2H, benzyl CH_2_), 7.35–7.38 (m, 2H, meta to acetyl), 7.04–7.09 (m, 2H, ortho to acetyl); ^13^C NMR (75 MHz, CDCl_3_) δ 21.1 (acetyl CH_3_), 66.5 (benzyl CH_2_), 121.7 (ortho C to acetyl), 128.7 (meta C to acetyl), 133.9 (para C to acetyl) 150.6 (Acetoxy-C), 156.3 (carbonyl C chloroformate), 171.2 (acetyl carbonyl C).

#### 6.2.31. 2′,3′-di-*O*-TBDMS-5′-*O*-[N^α^-(p-acetoxybenzyloxycarbonyl)leucyl]sulfamoyl adenosine (17)

The synthesis was performed like compound **13,** instead of p-nitrobenzyl chloroformate we used **16** as a reactant. The compound was obtained as a white solid in 62% yield. HRMS [ESI] *m/z*: calcd. for C_38_H_60_N_7_O_11_S_1_Si_2_ ([M-H]^-^) 878.3615, found: 878.3627.

#### 6.2.32. 5′-*O*-[N^α^-(p-acetyloxybenzyloxycarbonyl)leucyl]sulfamoyl adenosine (18)

The synthesis was performed like compound **14,** and the compound was obtained as a white solid in 28% yield. ^1^H NMR (300 MHz, MeOD) δ 0.92-1.29 (9H, m, Leu-δ’-CH_3_, Leu-δ″-CH_3_, Leu-γ-CH, Leu-ß-CH_2_), 2.25 (3H, s, CH_3_-Acetyl), 4.07–4.57 (6H, m, H_5_″, H_5_’, H_4_’, H_3_’, H_2_’, Leu-α-CH), 5.02–5.05 (2H, m, Bn CH_2_), 6.07-6.08 (1H, d, H_1_’, *J* = 6.67), 7.02-7.03 (2H, d, o-2CH, *J* = 8.10), 7.35–7.36 (2H, d, m-2CH, *J* = 9.06), 8.19 (1H, s, H_8_), 8.51 (1H, s, H_2_). ^13^C NMR (300 MHz, MeOD), 20.9 (CH_3_-Ac), 22.00 (Leu-δ’-CH_3_), 23.7 (Leu-δ″-CH_3_), 26.1 (Leu-γ-CH), 43.1 (Leu-ß-CH_2_), 57.2 (Leu-α-CH), 66.7 (C-5′), 69.2 (Bn), 72.3 (C-3′), 76.1 (C-2′), 84.5 (C-4′), 89.0 (C-1′), 120.1 (C-5), 122.7 (m-CH), 129.8 (o-CH), 136.1 (ipso-CH), 141.2 (C-8), 150.9 (C-4), 151.8 (p-CH), 153.8 (C-2), 157.2 (C-6), 158.5 (NH-COO) 171.2 (CO-Ac), 181.4 (CONH). HRMS [ESI] m/z: calcd. for C_26_H_34_N_7_O_11_S_1_ ([M+H]^+^) 652.2031, found: 652.2043.

### 6.3. In Vitro Inhibitory Activity Determination with Purified E. coli IleRS

The cloning, expression, and purification of *E. coli* IleRS was performed as described before^33^. To examine the inhibitory effect of the various compounds, the purified *E. coli* IleRS was used. Briefly, 10 nM IleRS, in 20 mM Tris, 100 mM KCl, 10 mM MgCl_2_, 5 mM β-mercaptoethanol, pH 7.5 was preincubated with the compound, at different concentrations, at 37 °C in the presence of 50 µM of the 14C-labeled isoleucine (Perkin–Elmer), 2 mg/mL tRNA (Sigma) and 0.5 mg/mL inorganic pyrophosphatase. After 10 min, pre-warmed ATP was added to the mixture at a final concentration of 500 µM. The reaction was quenched by the addition of 4 µL of quenching buffer containing 0.2 M sodium acetate pH 4, 0.1% *N*-lauroylsarcosine and 5 mM unlabeled isoleucine. 20 µL was spotted on 3MM Whatman paper. After thorough washing with cold 10% TCA, the filters were washed twice with acetone and air dried. Addition of scintillation liquid was followed by measurement of the radioactivity using the scintillation counter. The linear zone of enzyme activity was determined for each aaRS. The quenching time was picked within this zone at which approximately 50% of total RNA is aminoacylated. The quenching time of six minutes was used.

### 6.4. Time-Dependent in vitro Inhibitory Activity with E. coli Cellular Extract

To determine the time-dependent inhibitory activity of prodrugs (compound **14** and **18**), a mixture of inhibitor (at a stock concentration of 5 µM): S30 extract (1:4) was incubated at 37 °C for the specified period of time. The S30 extract was prepared as disclosed before^20^. The addition of inhibitor to cellular extract was done at time point zero and after 2, 15, 60 and 120 min, respectively, 5 µL of this mixture was added to 15 µL of the aminoacylation mixture which was kept at 37 °C and which contains phosphate (50 mM, pH 7.5), DTT (1 mM), *E. coli* MRE 600 tRNA (5 g/L purchased from Sigma), ATP (3 mM), magnesium acetate (10mM), potassium acetate (100mM), and 28.6 µM of 14C-radiolabeled leucine. The aminoacylation reaction was quenched after one minute by addition of 4 µL mixture of 0.2 M sodium acetate pH 4, 0.1% *N*-lauroylsarcosine, and 5 mM leucine. Then 10 µL of the reaction mixture was spotted on 3MM Whatman paper and this was transferred to 10% cold TCA solution. The papers were washed thoroughly with 10% cold TCA (twice), then the papers were washed twice with acetone and later dried in air. Dried papers were transferred to scintillation vial followed by the addition of scintillation liquid (12 mL), the amount of radionuclide incorporation was determined using a Tri-card 2300 TR liquid scintillation counter.

### 6.5. Antimicrobial Testing

The respective microbes were inoculated overnight in LB medium (5 mL) and cultured again in the next morning in fresh LB medium (5 mL) till it reached the OD_600_ of approximately 0.9. An amount of 10 µL of compounds made up in 50:50 DMSO:H_2_O solution was used for testing. Compounds were serially diluted using 50:50 DMSO:H_2_O mixture in a 96-well plate and 50:50 DMSO:H_2_O solution was used as control. Next, 90 mL of bacterial cell culture grown to a OD600 of 0.05 was added. The cultures were next placed in an incubator at 37 °C, and subsequently, the OD600 was determined after 20–24 h. Bacterial strains and fungi used for the evaluations are *S. aureus* ATCC 6538P, *S. epidermidis* RP62A, *E. coli* NCIB 8743, *P. aeruginosa* PAO1, *S. lutea* ATCC9341 and *C. albicans* CO11. All experiments were performed in triplicate.

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
