# Peer review of "Synthesis and Biological Evaluation of Lipophilic Nucleoside Analogues as Inhibitors of Aminoacyl-tRNA Synthetases"

_antibiotics, 2019, doi:10.3390/antibiotics8040180_

Round 1

Reviewer 1 Report

Aminoacyl-tRNA synthetases (aaRS) are a family of enzymes aminoacylating tRNA and thus involved in the translation of genetic information; they are essential for all organisms. Aminoacylation of tRNA involves the activation of cognate amino acids by coupling them to AMP to produce aminoacyl-AMP, from which the amino acid moiety is transferred to the 3’ end of tRNA. In the present study, the authors attempted to develop aaSA, a non-hydrolytic analogue of aminoacyl-AMP, as an antimicrobial agent. Since aaSA does not penetrate bacterial membrane, they modified it by two approaches for improving its permeability. None of the developed compounds exhibited the antibacterial effects that the authors expected to see, although the molecules showed sufficient in vitro inhibitory effects on IleRS and LeuRS, the targeted aminoacyl-tRNA synthetases. Thus, this study is not very successful from the view point of antibiotic development. What the reader can learn from this study is how to chemically synthesize the modified aaSA molecules and the data of the in vitro and bacterial experiments on them. These pieces of information would still be useful for researchers in the field.

Let me raise a few concerns that should be cleared before publication.

1.     The analogues of aminoacyl-AMP can act on not only bacterial aaRS but also human aaRS, and are therefore harmful, seemingly no good candidates as antibiotics. The authors mentioned mupirocin (l. 327), which selectively inhibits bacterial IleRS, harmless for humans. However, given the chemical structure of mupirocin, which is very different from aaSA, it is not very likely that aaSA can easily be developed into a selective inhibitor as mupirocin. The drawback of the authors’ approach seems to be lacking a feasible strategy for developing bacteria-selective aaSAs.  

2.     The second half of the abstract, which is the summary of the results, is not well organized, lacking important pieces of information, such as which E. coli aaRS they addressed, which types of compounds “6c” and “6d” stand for, and that the prodrug-type aaSAs had little effect on bacteria.

3.     There is no discussion about the result of 3.2.4 “Computational analysis of molecular properties”. The authors should discuss how these computational results are relevant to the biochemical and biological effects of the developed aaSAs.

4.     I feel that the title is too specific.  Not everyone in this field probably knows aminoacyl-sulfamoyl adenosine. The title should show that aminoacyl-tRNA synthetases are involved in the aim of this study.

Minor points: 

(l. 24 and in the title) “L” of “L-aminoacyl” should be a small capital.  

(Fig 1a) The linkage between the amino acid and phosphate moieties in aa-AMP is wrong. It should be “O”, not “NH”. I do not know if “NH” at the corresponding part in aaSA is correct. Please check it.

 (l. 109; l. 131) “ileRS” and “leuRS” are the abbreviations of isoleucyl-tRNA synthetase and leucyl-tRNA synthetase, respectively. These original names should be described, and "IleRS" and "LeuRS" are common as their abbreviations.

Author Response

Response to Reviewer 1 Comments

Point 1: The analogues of aminoacyl-AMP can act on not only bacterial aaRS but also human aaRS, and are therefore harmful, seemingly no good candidates as antibiotics. The authors mentioned mupirocin (l. 327), which selectively inhibits bacterial IleRS, harmless for humans. However, given the chemical structure of mupirocin, which is very different from aaSA, it is not very likely that aaSA can easily be developed into a selective inhibitor as mupirocin. The drawback of the authors’ approach seems to be lacking a feasible strategy for developing bacteria-selective aaSAs.  

Response 1: The authors are well aware that two main issues remain with this aaSA class of compounds, which is the uptake problem that is tackled here, and the bacterial selectivity. In fact, the latter part briefly has been dealt with already in this paper. The choice for a nitrobenzyl moiety is partially dictated by the need for a specific reductase, not available in mammalian cells, which this way can provide the required selectivity. Discussion on the selectivity issue was already present in the paper and now has been elaborated a bit more explicitly, and can be found in the paragraphs at lines 133-137 and 314-332.

Point 2: The second half of the abstract, which is the summary of the results, is not well organized, lacking important pieces of information, such as which E. coli aaRS they addressed, which types of compounds “6c” and “6d” stand for, and that the prodrug-type aaSAs had little effect on bacteria.

Response 2: The abstract has been enlarged a bit and now includes information on the additional suggested items and findings. Reference to the compounds has been clarified by including the alkyl group in the name (octyl and dodecyl chain, respectively), and correcting the assigned numbers to 6b and 6c, respectively.

Point 3: There is no discussion about the result of 3.2.4 “Computational analysis of molecular properties”. The authors should discuss how these computational results are relevant to the biochemical and biological effects of the developed aaSAs.

Response 3: We thank the referee for pointing to our failure to explain this table. We firstly reorganized and shortened the table, omitting the hydrogen bond donor and acceptor values, as our modifications do not bring much change. These values have been added to the result section itself. In the discussion part we now elaborated further on the obtained logP values in an additional paragraph. Two more literature references support this section.      

Point 4: I feel that the title is too specific.  Not everyone in this field probably knows aminoacyl-sulfamoyl adenosine. The title should show that aminoacyl-tRNA synthetases are involved in the aim of this study.

Response 4: We thank the referee for this suggestion. The title has been changed, removing the chemical name and pointing to inhibition of the aaRS, which indeed is more appealing for a broad public. The new title now reads “Synthesis and biological evaluation of lipophilic nucleoside analogues as inhibitors of aminoacyl-tRNA synthetases

Point 5: Minor points

(l. 24 and in the title) “L” of “L-aminoacyl” should be a small capital.  

      This typographical error has been corrected

(Fig 1a) The linkage between the amino acid and phosphate moieties in aa-AMP is wrong. It should be “O”, not “NH”. I do not know if “NH” at the corresponding part in aaSA is correct. Please check it.

      The figure has been updated. The “NH” assignment at the aaSA part is correct.

 (l. 109; l. 131) “ileRS” and “leuRS” are the abbreviations of isoleucyl-tRNA synthetase and leucyl-tRNA synthetase, respectively. These original names should be described, and "IleRS" and "LeuRS" are common as their abbreviations.

     These errors have been corrected throughout the whole manuscript.

Reviewer 2 Report

The paper demonstrates an interesting study around the SAR analysis of aaSA compounds and shows nicely how modification of the adenosine N6 position gives rise to potent in-vitro inhibition. Disappointingly this doesn’t translate in-vivo however, the results of the prodrug work are noteworthy and could be explored further. This article I would recommend publication following editing which is highlighted below and in the attached manuscript

Whilst the biological results highlight the utility of the compounds, the results for the prodrugs are less so. Specifically, the in-vitro assay generates excellent inhibitory results (Fig. 2) however, in contrast the in-vivo results are disappointing. I believe this is a consequence of the premature release of the drug from its pro-form (nitrobenzyloxy carbonyl is an excellent leaving group) because of chemical hydrolysis. The authors should determine hydrolysis rates in the presence of purified enzyme activators and chemical hydrolysis. Minor edits to the text layout and descriptors in the synthesis (Page 6, line 191-193) Bond angles in the Schemes are incorrect and should be modified.

Author Response

Response to Reviewer 2 Comments

Point 1: The paper demonstrates an interesting study around the SAR analysis of aaSA compounds and shows nicely how modification of the adenosine N6 position gives rise to potent in-vitro inhibition. Disappointingly this doesn’t translate in-vivo however, the results of the prodrug work are noteworthy and could be explored further. This article I would recommend publication following editing which is highlighted below and in the attached manuscript.

Response 1: We thank the referee for carefully reading the manuscript and following up all suggested rephrasing and correction of typographical errors have been implemented.

Point 2: Whilst the biological results highlight the utility of the compounds, the results for the prodrugs are less so. Specifically, the in-vitro assay generates excellent inhibitory results
(Fig. 2) however, in contrast the in-vivo results are disappointing. I believe this is a consequence of the premature release of the drug from its pro-form (nitrobenzyloxy carbonyl is an excellent leaving group) because of chemical hydrolysis. The authors should determine hydrolysis rates in the presence of purified enzyme activators and chemical hydrolysis.

Response 2: We do not agree with the referee regarding spontaneous hydrolysis of the nitrobenzyloxy carbonyl moiety. In fact, and as already discussed in the manuscript, this pro-moiety has been used before without reports on preliminary hydrolysis. Indeed, this prodrug strategy has been used for cancer treatment in animals. The selective hydrolysis at the point of intervention was accomplished, and was mediated by selective gene delivery for the site-selective reduction of the nitro moiety in mammalian cells. If the compounds would have been prone to spontaneous hydrolysis, selective targeting could not have been accomplished (see reference 29 of the manuscript). We hence are convinced the inhibitory effects as noted in cellular extracts in this paper are the result of rapid release of the active moiety by enzymatic cleavage of the promoiety. 

Point 3: Minor edits to the text layout and descriptors in the synthesis (Page 6, line 191-193) Bond angles in the Schemes are incorrect and should be modified.

Response 3: All suggested rephrasing and correction of typographical errors have been implemented (see also above at point 1).
All schemes have been corrected and redrawn and all bond angles have been corrected.
The missing structures 8, 9 and 10 have been added to scheme 3 for clarity instead of referring to our own literature.